# Application of Ultrasound Combined with Microbubbles for Cancer Therapy

**DOI:** 10.3390/ijms23084393

**Published:** 2022-04-15

**Authors:** Deepa Sharma, Kai Xuan Leong, Gregory J. Czarnota

**Affiliations:** 1Physical Sciences, Sunnybrook Health Sciences Centre, Toronto, ON M4N 3M5, Canada; kaixuan.leong@sunnybrook.ca (K.X.L.); gregory.czarnota@sunnybrook.ca (G.J.C.); 2Department of Radiation Oncology, Sunnybrook Health Sciences Centre, Toronto, ON M4N 3M5, Canada; 3Departments of Medical Biophysics, and Radiation Oncology, University of Toronto, Toronto, ON M4N 3M5, Canada

**Keywords:** cancer, cell death, vascular damage, ultrasound-stimulated microbubbles

## Abstract

At present, cancer is one of the leading causes of death worldwide. Treatment failure remains one of the prime hurdles in cancer treatment due to the metastatic nature of cancer. Techniques have been developed to hinder the growth of tumours or at least to stop the metastasis process. In recent years, ultrasound therapy combined with microbubbles has gained immense success in cancer treatment. Ultrasound-stimulated microbubbles (USMB) combined with other cancer treatments including radiation therapy, chemotherapy or immunotherapy has demonstrated potential improved outcomes in various in vitro and in vivo studies. Studies have shown that low dose radiation administered with USMB can have similar effects as high dose radiation therapy. In addition, the use of USMB in conjunction with radiotherapy or chemotherapy can minimize the toxicity of high dose radiation or chemotherapeutic drugs, respectively. In this review, we discuss the biophysical properties of USMB treatment and its applicability in cancer therapy. In particular, we highlight important preclinical and early clinical findings that demonstrate the antitumour effect combining USMB and other cancer treatment modalities (radiotherapy and chemotherapy). Our review mainly focuses on the tumour vascular effects mediated by USMB and these cancer therapies. We also discuss several current limitations, in addition to ongoing and future efforts for applying USMB in cancer treatment.

## 1. Introduction

In today’s world, imaging technology has revolutionized the way cancer is being diagnosed and treated. Techniques such as computed tomography (CT), positron emission tomography (PET), magnetic resonance imaging (MRI), ultrasound, and X-ray imaging provides detail of the body’s interior that allow clinicians and scientists to conduct in depth medical analyses. However, implementing these techniques in patients is not always feasible due to the costly nature of these technologies in general and severe health risk factor associated with them. Among all the above-mentioned techniques, ultrasound technology is known to be one of the safest and cost-effective choices for patients with severe health conditions. It is a non-invasive, portable, and radiation-free imaging modality.

Contrast agents such as microbubbles (MB) combined with ultrasound have acclaimed a huge success in recent years. These have been used both for diagnostic as well as for therapeutic purposes [1,2,3,4,5]. MB are gas-filled microscopic bubbles ranging < 10 μm in diameter. The gas core consists of either air, nitrogen, or gases such as perfluorocarbon, or sulfur hexafluoride stabilized by a shell made up of lipid, proteins, polymers, or phospholipids. The gas interior has a high molecular weight and low solubility [6,7]. The physical properties of MB are known to be dependent on the composition of the shell and gas core. MB remain inert under stable conditions and pass through the body, eventually decaying, with a half-life of less than ten minutes [6,7]. At present four different types of MB are commercially available and used for medical purposes including Albunex, Levovist, Optison, and Definity. In-depth detail of each MB composition and properties is discussed elsewhere [8,9,10,11,12]. When MB are exposed to ultrasound waves, MB start oscillating with expansions, and contractions. Ultrasound causes this MB cavitation with outcomes dependent on pressures applied. Stable cavitation is induced upon exposure to low acoustic pressure and causes a flow of fluid around the bubbles—a phenomenon known as microstreaming [13]. Inertial cavitation is characterized by oscillations of bubbles under high acoustic pressure causing bubbles to contract and expand to as much as double their size resulting in their fracture and collapse. During this process bubbles fragment into multiple tiny bubbles [14,15]. Cavitation of MB in close vicinity of cells or blood vessels causes several bioeffects on tissues or surrounding microenvironment depending on pressures and bubble exposure parameters [6]. MB in combination with ultrasound have been demonstrated as a new promising strategy for enhancing cancer therapies [16,17,18,19,20].

In recent years, the combination of ultrasound-stimulated microbubbles (USMB) with radiotherapy or chemotherapy has shown marked progress in treating various cancer types in preclinical studies [21,22,23,24,25]. USMB combined with radiotherapy allows the administration of low radiation doses that can elicit a similar effect observed when using single high doses of radiation alone [23] due to the activation of specific genetic pathways involving primarily endothelial cells [26]. Similarly, the risk of chemotherapeutic drug toxicity is limited when combining USMB with chemotherapy permitting the use of lower drug concentrations [1,27]. The underlying mechanism of combining USMB with radiotherapy might be different from using USMB with chemotherapy. However, the phenomenon of cavitation is utilized by both combined modalities enhancing endothelial vasculature permeability and/or causing vascular disruption and a process called sonoporation. Treatment results in shear stresses around the surrounding tissues that disrupts endothelial cells lining blood vessels resulting in an enhanced radio-or chemotherapeutic response. The process of sonoporation allows the temporary and reversible opening of endothelial vessel walls [28]. Sonoporation also creates pores in the cell plasma membranes increasing blood vessel permeability and allowing the intake of therapeutic substances that are required to treat a targeted area [20]. The pores generated by microbubbles vary from nanometer to micrometer in size depending on the size of bubbles [29].

The technique of combining USMB with chemotherapy has been successfully implemented in various preclinical tumour types including brain, bladder, breast, cervix, and prostate [30]. Recent work has also targeted CNS-based tumours [31,32,33]. Even with emerging advanced techniques, treating brain tumours remains challenging due to the unreachability of various drugs/molecules to the targeted area [34,35,36,37]. A protective barrier lining across the blood vessels in the brain also known as the blood-brain barrier (BBB) denies the entry of useful substances resulting in treatment failure. However, the use of focused ultrasound (FUS) combined with MB has now overcome this challenge by allowing chemotherapeutic drugs and therapeutic substances to pass through the BBB. FUS + MB enhances the uptake of chemotherapeutic drug delivery to treatment sites with minimal or no side effects [38,39,40].

In this review, we discuss important findings from the combination of these modalities in various preclinical as well as clinical studies. We discuss the underlying vascular mechanism of combining USMB with radiotherapy and or chemotherapy for the treatment of cancer. The utility of USMB in the delivery of chemotherapeutics across the BBB will also be discussed. We will also highlight the current obstacles, limitations, and future perspectives of these treatment modalities in cancer treatment.

## 2. USMB and Radiotherapy

Radiotherapy (XRT) is crucial in cancer treatments for many different cancer types. The versatility of XRT in delivering precise and accurate doses to target sites allows its usage in a wide range of clinical situations. It can be used as a primary or secondary treatment modality as radiation is often combined with surgery as an adjuvant or neoadjuvant treatment. Current radiation delivery method utilizes lethal doses of radiation (>40 Gy) administered in more tolerable fractions (1.8–2 Gy) over several weeks until cumulative curative doses are delivered to the target site [41]. Direct action of ionizing radiation (IR) on cells has been well characterized and studied in great depth in the past. The conventional theory of radiation-induced tumour cell death attributes DNA damage to be the primary player. IR can cause a direct effect on DNA or an indirect one, in which DNA damage is mediated by the generation of reactive oxygen species (ROS) [42,43]. This damage comes in many forms including DNA base damage, DNA depolymerization, cross-linking and DNA backbone breakages [44]. When large numbers of these breaks accumulate within the cell, cells undergo several responses including DNA repair, transcriptional response, cell cycle arrest, or apoptosis [45]. Cells are naturally capable of repairing such DNA damage through the activation of the DNA damage repair (DDR) pathways [44,45,46]. As such, much research has gone into understanding the DNA repair mechanism to exploit weaknesses that make cancer treatment more effective. In particular, the genetic or pharmacological suppression of the ataxia-telangiectasia mutated (ATM) gene has recently been of great interest in enhancing radiotherapy as it is a crucial component in detecting DNA double-strand breaks [44,45,46,47]. Patients with ATM inactivation are associated with a positive XRT prognosis across several many solid tumour types [48,49,50]. In vitro and in vivo experiments utilizing ATM inhibitors show great promise in enhancing radiation treatments in a multitude of different tumour types [51,52,53].

An alternative theory of IR-induced cell death has proposed that tumour cell death is primarily mediated by tumour vasculature disruption. Both single high dose (>8–10 Gy) and the fractionated dose (1.8–3 Gy/fraction) are known to induce endothelial damage leading to vascular disruption and finally causing tumour cell death. However, the underlying mechanism of tumour cell death induced by low and high doses is different (Figure 1). The tumour vasculature is a crucial component in enabling tumour growth and survival. This fundamental infrastructure is required to sustain tumour development, as blood supply delivers nutrients and oxygen to maintain cellular function [54,55]. Endothelial cells line the inside of vessels and have many crucial functions, including controlling vascular permeability, maintaining vascular tone, and modulating the immune system [56]. Studies have shown that high doses of ionizing radiation (>10 Gy) are capable of inducing tumour vasculature damage, which causes enhanced levels of endothelial cell death [57,58,59,60]. Acid sphingomyelinase (ASMase) is known to play a crucial role in facilitating endothelial cell apoptosis. ASMase is a membrane-bound enzyme that metabolizes sphingomyelin to generate ceramide, a secondary messenger. Under quiescent conditions, ASMase is localized within lysosomal compartments and when cells experience XRT-induced oxidative stress, leading to extracellular Ca2+ influx, these compartments fuse with the plasma membrane, releasing ASMase and facilitating apoptosis [61]. Radiation studies on vascular gastrointestinal (GI) of murine models showed that ASMase-mediated endothelial apoptosis is the primary factor that leads to stem cell dysfunction. Treating ASMase +/+ and –/– mice with single high radiation doses demonstrated significant endothelial apoptosis only in wild type (WT) genotype compared to their counterpart. This effect increased exponentially with increasing doses (0, 8, 12, 14, 15 Gy). The process of endothelial damage was abrogated using the basic fibroblast growth factor (bFGF), a vascular endothelial growth factor. ASMase knockout in murine models resulted in radioresistance in the vasculature [57]. Several studies conducted by Fuks and Kolesnick demonstrate the importance of ASMase-ceramide signaling in facilitating XRT mediated endothelial response [57,58,62,63,64,65,66]. The initial findings by Garcia-Barros et al. showed that fibrosarcoma (MCA/129) and melanoma (B16F1) tumour mice model exposed to 15 Gy of radiation resulted in tumour size reduction within 48 h post-treatment and continued to regress for several more days. However, tumours grafted onto ASMase –/– mice showed a radioresistant phenotype with low levels of endothelial apoptosis (comparable to the untreated conditions) and unimpeded tumour growth [58]. This highlighted the importance of ASMase in facilitating endothelial apoptosis, which leads to greater tumour control. Previous studies have shown that ASMase activation is only initiated upon XRT dose exposure > 8 Gy suggesting doses below this are not sufficient enough to induce ASMase activation to induce vascular disruption [64]. As such the use of USMB has been explored to enhance radiation doses < 8 Gy by stimulating the activation of ASMase and eliciting a vascular response (Figure 1).

Endothelial cells are known to be the primary players in USMB-induced radiosensi-tization. USMB perturb the endothelial cell lining leading to activation of the ASMase-ceramide pathway. Sonoporation-induced upon USMB increases vascular permeability allowing the oscillating MB to cavitate and circulate within the blood vessel wall [67]. This results in radiosensitization of endothelial cells and vasculature disruption enhancing overall tumour response (Figure 2). USMB and radiation-induced tumour vascular effects observed in preclinical studies are listed in (Table 1A). Many preclinical studies have explored the radio-enhancing effects of USMB on a variety of cancer types including acute myeloid leukemia (AML) [68], breast cancer [25,68], bladder cancer [24], colorectal carcinoma [69], esophageal squamous cell carcinoma (ESCC) [70], fibrosarcoma [23,68], human umbilical vein endothelial cells (HUVEC) [26,68], melanoma [71], and prostate cancer [22,68]. These in vitro and in vivo studies provided an insight into the molecular mechanisms that facilitate the synergistic effect upon USMB and XRT which is known to be dependent on ceramide. Experiments with XRT + USMB with HUVEC showed an upregulation of several apoptosis and ceramide-associated cell death pathways (SMPD1, UGT8, COX6B1, Caspase 9 and MAP2K1) upon treatment with a single dose of USMB + 8 Gy. The combined USMB + 8 Gy resulted in elevated ceramide level with enhanced cell death by 95% compared to XRT alone with 67%. Thus, the study showed the importance of ceramide in facilitating XRT + USMB cell death [26]. The presence of ceramide persisted up to 24 h post-treatment with co-localization with mitochondria, indicating the importance of mitochondrial ceramide in apoptosis signaling. On contrary, inhibiting ceramide release using sphingosine-1-phosphate (S1P), fumonisin B1 or monensin made the cell resistant to apoptosis. These findings highlighted the crucial importance of ceramide in enhancing cell death following XRT + USMB treatments [72]. Initial findings linking the XRT + USMB mechanism of action to the ASMase-ceramide pathway, in particular, showed that HUVEC and ASMase +/+ astrocyte cells showed elevated levels of ceramide and significantly reduced survival in response to USMB stimulation in vitro. This, combined with single-dose radiation showed an enhanced effect and a decreased cell survival [68].

These findings have also been recapitulated in several in vivo studies. Studies conducted by Czarnota et al. showed that the effects of USMB could enhance doses as low as 2 Gy to induce a cell death effect comparable to that of a single 8 Gy dose in PC3 mouse models. Ceramide level was found to increase in correspondence to increased XRT single high dose or XRT combined with low XRT dose. When mice were treated with S1P, ceramide activity and cell death showed no enhancement with increasing MB and radiation doses with levels comparable to that of untreated mice [21]. This further enforces the idea that XRT + USMB treatments depend on the generation of ceramide to initiate the apoptosis-signaling cascade. These findings have been validated in many other studies using different xenograft tumour types [22,23]. Specifically, El Kaffas et al. conducted a study looking into the importance of ASMase in activating ceramide following USMB and XRT. WT mice exposed to 2 Gy or 8 Gy XRT treatments showed elevated levels of cell death and microvascular density loss when treated with USMB compared to the XRT only treatment, up to 72 h post-treatment. However, this effect was not observed in ASMase –/– mice and mice pre-treated with S1P. Additionally, the use of power Doppler ultrasound was able to observe reduced perfusion and vascularity within tumours treated with USMB + XRT, while S1P treated mice and ASMase –/– retained vascular signals, despite being exposed to USMB and XRT. A similar tumour response was observed irrespective of treatments consisting of USMB + 2 Gy or a single high dose of 8 Gy. The study found that USMB + 2 Gy generates ceramide content similar to that of a single high dose of 8 Gy, which can induce tumour vascular damage [23]. Overall, the study found that combining USMB with low XRT doses elicits the effect of a single high dose. Both ceramide and its metabolites are known to be involved in signaling processes to regulate cellular function. An extensive study was conducted using a PC3 model in vitro and in vivo to determine ceramide signaling in tumour vasculature damage. The study explored the importance and effect of UDP glycosyltransferase 8 (UGT8) signaling on ceramide metabolism. In these experiments UGT8 gene was up and downregulated in mice bearing PC3 tumours. UGT8 facilitates the transfer of UDP-galactose to ceramide. Enhanced cell death indicating (37 ± 14) was observed in downregulated tumours following USMB and 8 Gy. In contrast, control and upregulated groups depicted minimal cell death indicating (10.7 ± 4) and (2 ± 0.04), respectively. An increase in cell death in UGT8 down-regulated group was found to be associated with high levels of ceramide accumulation and greater vascular damage [77]. Collectively, these studies suggest ceramide as a major determinant in USMB-induced radiosensitivity.

Recently, it was shown that doses below 8 Gy combined with USMB are known to regulate tumour response via activation of DNA damage pathways. A study conducted with mice implanted with human malignant glioblastoma U87-MG demonstrated enhanced cell death and reduced vessel density upon exposure to USMB + 4 Gy. Tumour growth was also found to inhibit significantly in a combined group. In addition, evaluation of DNA damage protein revealed a significant reduction in phosphorylation expression of breast cancer gene 1 (BRCA1), checkpoint kinase 1 (CHK1), and P53, which are known to be involved in the activation of DNA damage repair mechanism. On contrary, the level of γH2AX protein within the combined treated tumour sections was increased significantly. Their study suggested that low dose XRT combined with USMB enhances tumour response via DNA damage of tumour cells [82]. An interesting finding by Deng et al. indicated that USMB enhances radiosensitivity by repressing the levels of angiotensin II (ANG II) and its receptors (AT1R). An in vitro and in vivo study utilizing human NPC (CNE-2) and HUVEC were used to study the impact of USMB and XRT on ANG II and AT1R. A combined treatment of USMB with 2 or 8 Gy resulted in increased cell death and reduced blood flow. Additionally, the expression of ANG II and AT1R was found to be significantly higher with USMB and XRT treatments. Inhibiting ANG II and AT1R using its antagonist sensitized the cells by hindering the angiogenesis process. This work suggests that USMB sensitizes the effect of radiotherapy by inhibiting the expression of ANG II and AT1R [79]. Another proposed mechanism by He et al. suggests that USMB enhances the radiosensitivity by inactivation of progesterone receptor membrane component 1 (PGRMC1)-mediated autophagy. The study was conducted with glioblastoma cells (GL261, U251) in vitro as well as with mice bearing orthotopic glioblastoma tumours. The addition of USMB to the radiation group (2, 4, 8, 10 Gy) caused a decrease in cell viability and increased cell death. Additionally, a combination of USMB and 2 Gy resulted in higher accumulation of autophagosome and inhibition of autophagic flux and fusion of autophagosome with lysosomes. The expression of PGRMC1 was found to be reduced significantly in USMB and XRT groups. Overexpression of PGRMC1 caused marked inhibition in USMB-induced radiosensitization of glioblastoma cells. While using PGRMC1 inhibitor, AG-205 resulted in reduced cell viability and an increase in cell death suggesting abolishment of PGRMC1-mediated autophagy allows USMB enhanced radiosensitivity of glioblastoma [95].

Taken together, it can be rationalized that USMB can enhance the radiosensitization effect by activating or inactivating various membrane lipids or proteins including ceramide or PGRMC1, or by causing direct damage to DNA within tumour cells or by targeting tumour angiogenesis.

When combining two different treatment modalities optimizing their permutation and optimal integration is extremely important. There have been very few studies that have examined the effects of various parameters of USMB and XRT combinatorial therapy. A study by Kim et al. examined the effect of various biophysical parameters of USMB and XRT on PC3 tumours. Effects of peak negative pressures (250, 570 and 750 kPa), MB concentrations (8, 80 and 1000 μL/kg), and radiation doses (0, 2 and 8 Gy) were investigated. The results show that increased MB concentrations and higher radiation doses can elicit enhanced cell death effects. However, MB concentration appears to have a smaller impact on cell death when the peak negative pressure increases, as cell death levels remained unchanged at pressures of 570 kPa and 750 kPa combined with radiation. Ceramide used as a marker for cellular disruption was found to increase at 250 kPa with increasing MB concentration and radiation dose. However, ceramide levels were reported to plateau at higher ultrasound pressures as MB concentration and radiation dose increased. This suggested that at lower pressure bursting of the bubbles doesn’t take place but shear stress is implied on cells enhancing significant bioeffects. Whereas the effect due to high pressure (>570 kPa) causes MB to collapse resulting in saturated effects. The importance of peak negative pressure emphasizes MB cavitation as a crucial mechanism in enhancing USMB-XRT mediated cell death **[76]**. An additional study conducted by Klein et al. observed that the sequence of USMB and 8 Gy delivery showed no significant difference in PC3 mouse tumour models. Treatments delivered with timing of up to 24 h (0, 3, 6, 12, 24 h) between USMB and XRT delivery, or vice versa demonstrated enhanced tumour response. A higher tumour effect was observed at a 6-h difference between USMB and XRT, however, reversing the sequence of USMB and XRT did not impact the tumour response [80].

All the aforementioned studies have only been experimented upon using single-dose XRT and could differ with multiple USMB and or fractionated XRT (fXRT) regimens. Studies have recently been conducted combining USMB and fXRT to determine long-term treatment efficacy. Doses of 24 Gy in 12 fractions over 3 weeks (2 Gy fractions) and 45 Gy in 15 fractions over 3 weeks (3 Gy fractions) were able to induce tumour growth delay and reduced proliferation rates in PC3 tumours in mouse xenograft models under both fractionated regimens and USMB combination treatments [21]. Rabbits bearing PC3 xenografts showed a response to daily treatment of 2 Gy, over 5 days per week for 3 weeks. Tumour response primarily began to show in the second to the third week of treatment compared to the untreated controls and USMB only cohorts. However, combination treatment showed an enhanced response starting from week one indicating an enhancement in cell kill, vascular disruption, and growth inhibition until the endpoint (week 3) [81].

Thus, these studies established a foundation in the use of USMB in enhancing XRT with clinically relevant doses. However, there are limitations to the applications of USMB, which must be considered including prolonged use and safety over several weeks, alongside an fXRT regimen. At the current stages of study, USMB shows great promise in enhancing XRT doses through inducing endothelial apoptosis. Overall, all these studies suggest a direct involvement of tumour vasculature in enhancing tumour response. Treatment with USMB and XRT causes activation of the ASMase-ceramide pathway, leading to enhanced tumour cell death and vascular damage. Potentially, USMB can be used in clinically relevant XRT regimens to lower overall dose exposure while maintaining high efficacy in tumour cure.

## 3. USMB and Chemotherapy

Chemotherapy incorporates the use of several drugs to destroy cancer cells. The process involves oral consumption, topical agents, or intravenous intake of chemotherapeutic drugs. All these methods use high doses of chemotherapeutic drugs and have proven to be effective in curing different types of malignant tumours. However, the side effect that comes along with it cannot be overlooked due to high drug toxicity. New techniques of drug delivery that require low drug concentration are needed to overcome this challenge. The combination of ultrasound and MB has shown increased cancer cell death by improving drug delivery and enhancing the efficacy of several drugs [86,88]. The requirement of low drug concentration while using USMB has demonstrated reduced toxic side effects [19].

USMB combined with chemotherapy can be administered in two different ways. The first way involves a co-injection of drugs and MB followed by an ultrasound application. The second method uses drug-loaded MB followed by an ultrasound exposure that allows the local release of drugs to the targeted site [6,96,97,98]. Schematic representations of both the methods are presented in (Figure 3). Depending on the low and high-frequency ultrasound waves, MB cavitation-induced sonoporation can easily deliver several therapeutic substances such as drugs and genes [20]. Alternatively, USMB facilitates the intracellular delivery of chemotherapeutic drugs through clathrin and flotillin-dependent endocytosis [99,100,101]. Clathrin-dependent endocytosis is mediated through the internalization of receptor-bound macromolecules and the process of flotillin-dependent endocytosis involves palmitoyltransferase aspartate–histidine–histidine–cysteine (DHHC)5 and the Src-family kinase Fyn regulation by USMB [100,101]. Recently a new technique that is widely used for transporting therapeutic agents to target sites is sonoprinting. The technique of sonoprinting utilizes nanoparticle-carrying MB that deposit the nanoparticle patches onto the cell membranes. Sonoprinting requires high ultrasound acoustic pressure and long pulses to deliver the nanoparticles effectively onto the cells [102].

## 4. Ultrasound-Stimulated Microbubbles-Mediated Drug Delivery

### 4.1. Co-Administration of Microbubbles and Chemotherapeutic Drugs

Several in vitro and in vivo studies have demonstrated the effectiveness of USMB in facilitating the delivery of chemotherapeutic drugs in a safe and feasible manner [86,88,103]. USMB combined with chemotherapy has shown substantial enhancement in intracellular drug uptake demonstrating several therapeutic benefits. A direct correlation between intracellular drug uptake and tumour cell death has been documented in various preclinical studies. An in vitro study carried out by Karshafian and colleagues investigated the efficacy of USMB and chemotherapy in breast cancer (MDA-MB-231) and prostate cancer (PC3) cells. Breast cancer cells exposed to docetaxel (DTX)(Taxotere^®^) followed by USMB administration demonstrated a decrease in cell viability from 65% to 7% using a drug concentration of 0.0001 nmol/mL and 0.05 nmol/mL, respectively. DTX cytotoxicity was enhanced in PC3 cells as well however MDA-MB-231 cells were found to be more sensitive to the drug [103]. Another study found similar results using retinoblastoma Y79 cells. A decrease in cell viability of 34.9% was reported with a combination of USMB and doxorubicin (DOX) (1 μM). However, DOX group alone caused a decrease of 50.9% within 48 h [104]. A study conducted by Escoffre et al. utilized Vevo Micromarker microbubble-assisted ultrasound to evaluate the efficacy of DOX delivery (5 μM) in human glioblastoma astrocytoma cells (U-87 MG). A reduction in cell viability by 2.5-fold following USMB and DOX was reported within 48 h in U-87 MG cells [105]. A study by Lammertink et al. demonstrated that exposing head and neck cancer cell lines to USMB resulted in a 2.7-fold higher uptake of cisplatin. The intracellular accumulation of cisplatin was found to be concomitant with an increase in DNA double-strand breaks by 82% [106]. A study conducted with BxPC3 pancreatic cancer cells treated with a combination of USMB and gemcitabine (1 nM–1 mM) resulted in cell viability decrease at 48 h. However, the study found no enhancement in the uptake of gemcitabine drug followed by a combined treatment of USMB and chemotherapy [107]. This suggests that USMB facilitated intracellular drug uptake might be dependent on the type of chemotherapeutic drugs or the type of cell line models.

An in vitro and in vivo study by Heath et al. investigated the efficacy of USMB and chemotherapy (cisplatin and cetuximab). This study used four head and neck cancer cell lines including SCC-1, SCC-5, Cal27, and FaDu for in vitro work, and for in vivo work, tumours were induced by injecting SCC-5 cells in athymic female nude mice. The result from the in vitro study showed cell death to significantly increase in the combined treated group of UMSB and cisplatin or USMB and cetuximab. Furthermore, the in vivo experiments demonstrated that the tumour size significantly reduced following combined treatment of USMB and cisplatin or USMB and cetuximab [108]. Similarly, a study by Zhao et al. showed epirubicin hydrochloride (EPI) (20 mg/kg) combined with USMB induced tumour growth inhibition in male nude mice bearing HL-60 tumours. The application of ultrasound caused substantial uptake of EPI by tumour cells [109]. Chemotherapeutic drugs are known to enter and accumulate in cancer cells for a longer period of time even after the diminishment of the cavitation effect [110]. Many studies have reported a higher accumulation of chemotherapeutic drugs around the tumour blood vessels area, indicating that chemotherapy exerts a direct impact within or around tumour vasculature. Ultrasound, MB and chemotherapy-induced tumour vascular effects observed in preclinical studies are listed in (Table 1B). The overall tumour response mediated by USMB and chemotherapy is mainly known to be dependent on tumour blood perfusion. Many groups have established a direct correlation between tumour vascular damage leading to tumour growth inhibition and better survival of animals following USMB and chemotherapy. A study reported by Goertz et al. incorporated USMB with DTX (5 mg/kg) to examine the efficacy of this combined treatment on the PC3 xenograft mouse model. Diminishment in blood perfusion and significant cell death was demonstrated within 24 h of combined treatment. Animal exposed to a combined treatment of USMB-DTX further showed tumour growth inhibition at 4–6 weeks compared to a group that remained untreated or treated with individual treatment alone. The study found a strong correlation that existed between tumour perfusion and cell death observed acutely at 24 h with that of tumour shrinkage/inhibition observed after several weeks [86]. A similar study was conducted using USMB and metronomic cyclophosphamide (MCTX) (20 mg/kg/day). The breast xenograft model (MDA-MB-231) exposed to combined treatment demonstrated acute depletion in tumour perfusion and significant cell death followed by inhibition of tumour growth observed longitudinally. No such observations were reported for control tumours [88]. The phenomenon of tumour cell death following USMB and chemotherapy is known to be dependent on the regulation of Bcl-2-associated X protein (Bax) and B-cell lymphoma protein 2 (Bcl-2) protein expression. A study by Shen et al. demonstrated that nude mice bearing hepatic carcinoma upon exposure to USMB combined with chemotherapy (cisplatin, mitomycin, and 5-fluorouracil) resulted in disruption of microvessel walls and enhanced apoptosis indicating a higher level of Bax and lower level of Bcl-2 protein expression. Furthermore, the tumour size in combined treated mice was significantly smaller compared to the pre-treatment group [111]. Similar studies were also conducted in New Zealand white rabbits bearing VX2 tumours. The study utilized low-intensity ultrasound irradiation and MB combined with DOX (6 mg/kg). An increase in the tumour perfusion was reported immediately following combined treatment with an accumulation of DOX mainly found around the blood vessels area. In addition, tumour growth inhibition following USMB and DOX was also reported. It has to be noted that the tumour perfusion here is reported to increase with USMB and chemotherapy in contrast to what is observed in previously mentioned studies. However, in both cases, the uptake of chemotherapeutic drugs is significantly increased resulting in overall tumour response [91,92]. The enhanced delivery of chemotherapeutic drugs following USMB is known to be a result of declination in the interstitial fluid pressure (IFP) within the tumour [92].

### 4.2. Drug-Loaded Microbubbles Combined with Ultrasound

Drug-loaded MB combined with ultrasound have proven to be an efficient method for the delivery of various chemotherapeutic drugs. Loading of drugs onto the MB can be achieved in various ways. It can be done by dissolving the drugs in the oil layer inside the MB shell or by incorporating the drugs in the MB shell or by attaching the drugs to the surface of the MB [6,98]. Once the drug-loaded MB reach their target location high acoustic ultrasound pressure is applied to allow the local release of drugs that is caused by ultrasound-targeted microbubbles destruction (UTMD). The duration of this entire process of MB destruction is known to be dependent on MB size and its shell’s thickness [112]. Studies have found the drug-loaded MB method to be more advantageous compared to the co-administration of drugs and MB. One of the benefits of this method includes assurance of drug release at the targeted site and not at the unwanted area. This method is known to increase the accumulation of various drugs around the blood vessels area as a result of sonoporation that takes place during UTMD. Drug concentrated around the blood vessel region enhances tumour response which might be useful when using this method. Loading the drugs onto the MB assures complete drug protection allowing minimal or no degradation of drugs. This allows the usage of low drug concentrations. Additionally, this method allows monitoring of the movement of drug-loaded MB that can be achieved using low acoustic pressure and once the drug reaches its desired site, MB can be burst using high acoustic pressure [98].

Several studies have incorporated the method of drug-loaded MB followed by ultrasound exposure to determine the efficacy of chemotherapy. Tinkov et al. performed both in vitro and in vivo experiments using DOX-loaded phospholipid MB and ultrasound. Their in vitro work utilized 293/KDR cell line exposed to DOX-loaded MB followed with or without ultrasound exposure. Results indicated inhibited proliferative activity in cells treated with DOX-loaded MB and ultrasound by 3-fold compared to untreated or DOX-loaded MB only [113]. In vivo work using rats bearing pancreatic carcinomas revealed a 12-fold higher uptake of a drug as a result of which significant tumour growth inhibition was observed in the group receiving DOX-loaded MB and ultrasound compared to control groups [114]. Similarly, Hou et al. used BEL-7402 cells that were treated with 10-hydroxycamptothecin (HCPT) loaded polylactic acid (PLA) MB following exposure to ultrasound. Results demonstrated higher intracellular drug uptake and significant cell growth inhibition in HCPT-loaded PLA MB combined with the exposure to ultrasound compared to untreated cells or HCPT-loaded PLA MB without exposure to ultrasound [115]. A study by Escoffre et al. used human glioblastoma cells (U-87 MG) exposed to DOX liposome-loaded MB followed by ultrasound exposure. An increase in DOX uptake and decrement in cell viability by 4-fold was observed in the combined treated group compared to drug treatment alone [116]. A study by Lai et al. used BGC-823 gastric cancer cells to examine the effect of DTX-loaded lipid microbubble (DLLD) and ultrasound. It was found that the cell viability was significantly inhibited at 48 h in the group that was exposed to drug-loaded MB and ultrasound compared to the group that received the drug treatment alone. Additionally, the cells in the S-phase were significantly lower while that in the G2/M phase increased in the drug-loaded MB and ultrasound group indicating more cells undergoing apoptosis following the combined treatment. Furthermore, the combined treated group induced higher Bax and lower Bcl-2 expression [117]. Many other studies have also reported Bax and Bcl-2 to be crucial factors for modulating tumour response when using drug-loaded MB combined with ultrasound [118]. Along with Bax and Bcl-2, direct damage to tumour vasculature is also known to impact tumour response significantly when using USMB and chemotherapy. A study by Lin et al. examine the effect of USMB and pegylated liposomal doxorubicin (PLD) (6 mg/kg) on mouse-bearing colorectal adenocarcinoma CT-26 tumour. A combined treatment of USMB and PLD induced significant tumour growth inhibition along with a significant reduction in tumour blood perfusion. In addition, the histological examinations of tissue samples from the combined treated group revealed abnormal tissue morphology with damaged tumour blood vessels. This suggested that vessel disruption can increase the uptake of chemotherapeutic drugs resulting in the hindrance of tumour growth [87].

Other in vivo work has also been carried out using the technique of drug-loaded MB and ultrasound. Kang et al. studied the effect of DTX-loaded lipid MB combined with ultrasound on VX2 rabbit liver tumours. The treatment was administered thrice on days 1, 4, and 7. Results demonstrated a significant reduction in the tumour size following a drug-loaded MB and ultrasound treatment. Further analysis revealed a lower cell proliferation and higher apoptosis in the tumour section that received the identical treatment. Comparative to the drug-loaded MB group alone, the animal survived the longest with the treatment of drug-loaded MB and ultrasound [119]. An in vivo and in vitro work was performed by Hu et al. using human GRC-1 granulocyte renal carcinoma cell line and nude mice bearing tumour. For in vitro study, the cell survival rate was calculated, and for in vivo work, tumour growth was measured. Sunitinib-loaded MB with ultrasound treatment exhibited significant tumour growth inhibition both in vivo and in vitro [120]. Yu et al. incorporated a different approach by combining ultrasound-mediated paclitaxel (PTX)—and miR-34a-loaded MB. Their study evaluated the synergetic inhibitory effects of both these treatments using U14 mouse cervical cancer both in vivo and in vitro. Results from in vitro work demonstrated inhibition in cell proliferation and reduction in microvascular density. Additionally, Bcl-2 and cyclin dependent kinase 6 (CDK6) expression were also reported to diminish with a treatment consisting of ultrasound-mediated PTX-and miR-34a-loaded MB. Similar results were obtained with in vivo work. Significant inhibition in tumour size, microvessel density, and expressions of Bcl-2 and CDK6 was observed in mice bearing tumours. Thus, the study highlighted the importance of combining two different approach for achieving a synergistic tumouricidal effect [121].

The efficacy of chemotherapeutic drugs is dependent on several factors, one of which includes the type of drug used. Work by Cochran and colleagues studied the efficacy of the hydrophobic and hydrophilic drugs loaded onto the MB. This comparative study was performed in vitro and in vivo using DOX and PTX-loaded MB followed by ultrasound exposure. A comparison was made to see the efficacy of hydrophobic drug PTX and hydrophilic drug DOX. The drug delivery efficacy was based on the ability of an individual drug to penetrate through 400 nm pores. It was found that the delivery efficacy of hydrophobic drug PTX was 20 times (129.46 ± 1.80 μg PTX/mg) higher compared to hydrophilic drug DOX (6.2 μg/mg). Higher release of PTX overtime was reported upon ultrasound treatment. Greater penetration of drug-loaded MB across tumour vasculature of rabbits bearing VX2 liver tumour was also noted. Additionally, human breast cancer cell lines exposed to PTX-loaded MB and ultrasound demonstrated a significant reduction in cell viability at 72 h suggesting greater tumoricidal activity of PTX-loaded MB and ultrasound [122]. Another factor that affects the drug efficacy is the dosage of drug used. Recently, Delaney and colleagues determined the amount of drug entrapped into the MB that are required to cause tumour response using gemcitabine-loaded MB. In vitro work was carried out with MIA PaCa-2 cell lines and in vivo work was performed using MIA PaCa-2 PDAC murine model. Results demonstrated that a gemcitabine concentration ≥500 nM entrapped in a MB is required for cells to undergo complete cell death in vitro. In vivo results showed no significant inhibition in tumour growth using similar drug concentrations entrapped in MB. This suggests a higher concentration of drug might be required in vivo to acquire similar outcomes as observed in vitro [112].

Thus, the methods of co-administration of MB with chemotherapeutic drugs as well as drug-loaded MB following ultrasound exposure are suggested to be safe, feasible, and effective. Very few studies have performed a comparative analysis using both methods. One of which includes the study by Ruan et al. using CD1-nude mice bearing RT112 bladder tumour. Gemcitabine-conjugated MB and ultrasound combined with radiation therapy were found to be more effective compared to co-administration of MB and gemcitabine followed by ultrasound and radiation therapy in terms of delaying tumour growth longitudinally [123]. This suggested the drug-loaded MB method to be more effective than the co-administration one however, more comparative studies are needed to draw a significant conclusion.

## 5. Effect of USMB and Chemotherapy on 3D Cultures (Spheroids)

In recent years, the combination of USMB and chemotherapy has shown immense success in treating in vitro and in vivo cancer models. The delivery and efficacy rate of chemotherapeutics drugs following USMB has shown significant improved outcomes [19,124]. However, intratumoural heterogeneity has always been and is still a hindrance while treating cancer following these treatments [97]. The translation of preclinical work to clinical settings often faces several challenges due to the differences in the tumour microenvironment. Recently, the discovery of three-dimensional (3D) culture models has bridged the gap between in vivo work moving forward to in vivo studies and finally leading to clinical trials. Using this kind of model allows customization of microenvironments of several tumour types allowing personalized drugs required for specific tumour types. When cells are grown in 2D monolayer, a uniform proliferation is seen however, this is not realistic in the case of tumour cells in cancer patients. Therefore using a 3D model will result in changes in cell proliferation rate. Another advantage of using this model is that in 2D the phenotype of tumour cells are homogenous however, using 3D model will allow dealing with a heterogeneous phenotype of tumour cells which is a more realistic approach for treating cancer patients. Studies have shown that growing cells on 3D often results in drug treatment failure due to lower penetration of drugs on the target site resulting in drug resistance which has also been seen in a clinical setting [125,126]. Overcoming such limitations will help future patients with a high success rate in curing cancers. The physiology of cells in a 2D monolayer is known to be completely different from those grown in 3D. Variability in gene expression and behavior of different cell types within one microenvironment will allow targeting different signal transduction pathways. 3D tumour spheroid model allows monitoring the spatial distribution of drugs that can help predict the efficacy of drug delivery. Very few studies have utilized 3D cultures to examine the efficacy of USMB-induced drug delivery. A recent study found an enhanced penetration and significant uptake in the chemotherapeutic drug following USMB. A 3D tumour spheroid model (MDA-MB-231) upon exposure to USMB demonstrated an increase of DOX (50 μM) uptake by 1.2-fold [127]. A study by Roovers et al. examined the effect of sonoprinting DOX-liposome-loaded MB and ultrasound on tumour spheroids. A large deposition of DOX was found at the outer cell layers of the spheroids that caused cells to die significantly [128]. A comparative study was carried out to examine the effect of USMB and DOX in 2D (monolayer-cultured cells) and 3D cultures (spheroids) of A549 non-small cell lung cancer cell and MDA-MB-231 triple-negative breast cancer cell. Results demonstrated a higher drug penetration in the spheroids seen at the edge and middle zones. No such effect was seen in the 2D model. More specifically, A549 spheroids upon 20-sec exposure of ultrasound and MB plus 1-h incubation with DOX resulted in fluorescence intensity increased from 1.4- and 1.8-fold into the edge and middle zones, respectively. A similar result was obtained with MDA-MB-231 spheroids. Ultrasound exposure of 20-sec and 1-h incubation with DOX caused fluorescence intensity increased by 1.7- and 1.6-fold in the edge and middle zones, respectively. Thus, increasing the exposure and incubation time of USMB and DOX enhanced the fluorescence intensity in both the tumour spheroids. The accumulation of DOX into tumour spheroids was observed following low-intensity pulsed ultrasound in combination with MB. However, no drug enhancement was observed in monolayer-cultured A549 and MDA-MB-231 cells [129].

Data collected with 3D tumour spheroids provide more accurate and in-depth knowledge compared to the 2D cell cultures model. More studies are required with 3D tumour spheroids, as it resembles the spatial complexity of actual human tumours. Future studies should also incorporate the 3D cell culture systems developed using patient-derived cells or patient-derived organoids. This will provide a more realistic clinical approach that can be made highly personalized and will the targeting of different types of tumour cells in a single patient.

## 6. USMB-Mediated BBB Disruption for Targeted Drug Delivery

The blood-brain barrier (BBB) is a protective diffusion barrier comprised of joined endothelial cells lining the blood vessels in the brain. The barrier hinders the passage of non-essential harmful substances across the brain. Equivalently, it also denies the entry of medication and essential therapeutic substances that are used to treat several brain diseases. The traditional method of treating neurodegenerative disorder involves an invasive injection of pharmacological agents into the brain parenchyma or by disrupting the BBB via chemical means for the uptake of chemotherapeutic drugs. Both these method lead to impairment in brain functioning and its development [130]. Treating brain disorders safely with minimal or no damage to surrounding brain tissue has been a goal of scientists and neurologists across the globe. Several techniques have been developed and tested to overcome this challenge, one of which includes the use of ultrasound [33]. Ultrasound in BBB disruption and its opening has been reported much earlier. However, the use of ultrasound on its own requires a high acoustic power that causes a magnitude of neurological damages [131,132]. Later, it was shown that ultrasound used in conjugation with MB could compensate for the use of a larger amount of energy that is required for the disruption of BBB. Low-intensity ultrasound combined with MB demonstrated BBB opening and disruption with minimal or no damage to surrounding tissue [133,134]. In recent years, focused ultrasound-microbubble (FUS-MB)-mediated disruption of the BBB has grabbed immense attention. The technique is proven to be non-invasive, safe, and opens the BBB temporarily and reversibly [135,136]. FUS-MB-mediated BBB disruption is based on the cavitation process (discussed earlier). Ultrasound causes MB oscillations that exert pressure on the endothelium tight junction forcing it to move apart [39,137]. Studies suggest that FUS-MB-mediated disruption and opening of BBB last for up to a few hours to days [138]. Depending on the opening of BBB, molecules as small as 3 kDa or as large as 2000 kDa can cross the brain barrier. Stable cavitation is known to cause a relatively smaller opening of BBB (2.3–10.2 nm) whereas inertial cavitation leads to BBB opening up to (54.4 nm) [139].

Several preclinical studies by Treat et al. utilized MRI-guided focused ultrasound to examine the feasibility of chemotherapeutic drugs across the BBB. Their study investigated DOX accumulation across BBB using MRI-guided focused ultrasound with MB. The experiments were carried out on rat brains. The results demonstrated that using lower MB doses combined with FUS to disrupt the BBB resulted in 886 +/− 327 ng/g accumulation of DOX in the brain. Increasing the MB dose resulted in enhanced DOX concentrations up to 5366 +/− 659 ng/g however, with higher tissue damage observed across tumour tissue [140]. Their other study included rats bearing 9L gliosarcoma to study the effect of FUS-MB mediated DOX delivery on rat survival. An increase in survival of rats that received FUS-MB+DOX treatment was reported compared to rats that received no treatment or DOX treatment alone [141]. Many more studies were carried out in vivo to examine the efficacy of this technique on drug accumulation and animal survival. A study by Liu et al. examined the treatment efficacy of FUS-MB in delivering 1,3-bis(2-chloroethyl)-1-nitrosourea (BCNU) to the brain of rats bearing C6 glioma. It was found that FUS-MB caused drug penetration in tumour-implanted brains by 202% without causing any bleeding. Animal survival was found to significantly improve in the combined treated group compared to untreated control groups [142]. A similar study was recapitulated in a mouse model. Two mouse model-bearing glioblastomas (GL261 and SMA-560) were utilized to study the safety and feasibility of BBB opening and disruption following FUS-MB-mediated delivery of DOX. An increase in DOX concentration at the contralateral hemisphere was found with FUS-MB treatment. Additionally, enhanced animal survival with inhibited tumour progression was also observed [143]. A similar study was conducted by Liu et al. that determined the impact of temozolomide (TMZ) in nude mice. Mice bearing U87 human glioma were used to examine the effect of FUS-MB-mediated BBB disruption/opening and TMZ delivery in the brain. Results demonstrated a 2.7-fold increase (19 ng/mg) in TMZ accumulation in the animal brain receiving a treatment of FUS-MB. However, the animals that received a treatment of TMZ alone demonstrated a drug accumulation of 6.98 ng/mg only. Further, the combined FUS-MB+TMZ caused inhibition in tumour growth resulting in prolonged survival of animals [144]. A recent study by Englander and colleagues carried out with murine pontine glioma model demonstrated that the usage of FUS-MB in BBB opening and disruption could preserve the cardiorespiratory and motor function. The study further found that the hemorrhage and inflammation observed within a tumour region were similar between the treated and control group indicating that the FUS-MB had no additional adverse effect [136].

Several groups have also studied the effect of drug-loaded MB following ultrasound on BBB opening. A study by Ting et al. used C6 glioma bearing rats that were exposed to FUS and MB carrying BCNU. Results revealed a 5-fold higher circulatory half-life of chemotherapy drugs. Tumour progression was found to significantly inhibit with improved animal survival observed following BCNU-MB and FUS treatment [145]. A study by Fan et al. utilized an antiangiogenic-targeted strategy to examine the effect of BCNU-loaded MB with FUS to induce BBB opening for efficient delivery of therapeutic drugs. The result demonstrated the opening of BBB in a rat model safely and effectively resulting in enhanced uptake of BCNU leading to increased animal survival [146]. Another study by Fan et al. examined the effect of FUS and superparamagnetic iron oxides (SPIO)-conjugated, DOX-loaded MB on a rat model. DOX deposition within the tumour area was found to increase by 2.05-fold compared to remaining brain tissue [147].

Thus, all these studies provide an insight that FUS exposure combined with MB has the potential to safely, but transiently open the BBB and thus can increase the delivery of various drugs and therapeutic substances across the brain.

## 7. Clinical Applications of USMB

### 7.1. USMB and Radiotherapy

The unique properties of USMB allow for a great variety of applications in the clinical setting. However, given the mounting evidence, that preclinical studies show in enhancing radio and chemotherapy, clinical studies involving combined treatments have not yet been able to replicate these results in clinical settings. So far, there have mostly been studies focusing on the safety and feasibility of USMB for patient treatment.

A clinical study conducted by Eisenbrey and colleagues examine the efficacy of USMB in hepatocellular carcinoma patients with USMB and transarterial radioembolization (TARE). The total number of patients used in the study was 28; out of which 11 received TARE alone treatment and 17 received a combined treatment of TARE + USMB. Patients treated with TARE + USMB showed a greater response compared to patients undergoing TARE only, with a large percentage of patients showing partial or complete response to the combination treatment. USMB destruction in the combination treatment showed to be well tolerated with no notable side effects in a 4–6 month post-treatment evaluation. No compromise in liver function was observed with the addition of USMB treatments [148]. Despite the small patient cohort, it shows great promise in the safety of adding USMB to existing radiation protocols to enhance the curative effect.

Thus, this pilot study shows a proof of concept for the safety and efficacy of USMB treatments; however, there are still several limitations when translating the treatments from preclinical studies to clinical settings. Dose delivery to in vivo studies does not correlate one-on-one with human tolerated doses and while MB are used frequently as a clinical contrast agent, therapeutic concentrations must be adjusted to compensate for differences between patients and tumour characteristics. Ultrasound frequencies and peak negative pressures in the clinic can also differ in accordance to depth, tissue types, and organ location as ultrasound attenuation can affect the acoustic field affecting the MB, altering the efficacy. However, the use of FUS has promise in overcoming these architectural limitations, as the technology of therapeutic ultrasound advances. In addition to spatial limitations, biological limitations also pose an obstacle that must be taken into account. Given the proposed mechanism of tumour vasculature destruction as the primary mechanism of action, USMB efficacy will highly depend on a tumour’s capabilities to re-vascularize and overcome endogenous cell survival mechanisms such as the hypoxia-inducible factor (HIF-1) response [149]. As a result, USMB + XRT treatments are best conducted alongside vascular imaging modalities, such as power Doppler, MRI, or contrast-enhanced CT, to monitor tumour vascular changes and tumour progression. Further optimization of USMB + fXRT dosage and temporal treatment must be conducted to best align it with standard care practices in the clinic. However, despite these limitations, the use of USMB as a radiotherapy-enhancing agent shows great promise as a safe addition or alternative treatment to conventional XRT in clinical settings.

### 7.2. USMB and Chemotherapy

The combination of USMB and chemotherapy has undergone several clinical trials compared to XRT and has shown promising outcomes in enhancing existing drug therapies. Patients with inoperable pancreatic cancer receiving USMB and gemcitabine showed better treatment tolerability compared to control groups. The study reported no enhanced side effects when combining USMB with chemotherapy [150,151]. Further clinical studies looking into the safety of USMB alongside chemotherapy showed that minor adverse effects were observed in patients with advanced digestive system malignancies while providing enhanced chemotherapy sensitivity [152].

The blood-brain barrier (BBB) is a commonly known limiting factor in chemotherapy of glioblastoma [33] and other treatments of ailments of the brain [153,154]. The semi-permeable barrier across the brain prevents the entry of foreign substances from entering and potentially harming the neural tissue in the brain. As such, this limits drug delivery towards targeted diseased areas. In the clinical trial conducted by Carpentier and colleagues [33], the use of an implanted ultrasound device (SonoCloud System) was used to stimulate MB to enhance uptake of carboplatin in patients with glioblastoma. The preliminary results showed successful BBB disruption with no toxicities occurring as a result of USMB stimulation. However, the effect of tumour response and enhanced carboplatin uptake has yet to be reported. Given the highly toxic and broad range of effects that chemotherapy drugs have upon treatment, the importance of safety when adding new modalities alongside standard treatment drugs is crucial and cannot be overlooked. As such, understanding the mechanistic action of USMB on primary tumour masses and its microenvironment is critical for ensuring the safe application of enhancing drug targeting and localization.

## 8. Conclusions and Limitations

The present review focuses on the implementation of ultrasound and MB for the enhancement of radiation effects as well as for the delivery of chemotherapeutic drugs. Ultrasound exposure causes MB to cavitate, which results in enhanced tumour vascular permeability. This allows the entry of various drugs and therapeutic substances to the treatment site. Intercellular uptake of drugs by tumour cells results in their death resulting in overall tumour cure. Similarly, the use of low intensity focused ultrasound combined with cavitating MB allows a safe opening of BBB resulting in greater drugs uptake. This phenomenon has shown a significant improvement in treating various brain-related diseases. Studies have shown greater tumour response when USMB is combined with radiotherapy suggesting USMB to be a potential radiosensitizer. The risk of using a high radiation dose is reduced when USMB is conjugated with lower radiation doses thus eliciting similar therapeutic effects. Similarly, reduced drug toxicity and severe side effects are minimized when USMB is delivered combined with chemotherapeutic drugs. Thus, the safety and feasibility of USMB have translated its usage from preclinical to several clinical trials that are in progress. Despite the many benefits of ultrasound and MB, its application for therapeutic use still holds several unanswered questions. For instance, the short half-life of gas-filled MB might be a hindrance for its use in clinical settings. There is a need for stable biocompatible MB for patient use. More descriptive and comparative studies should be conducted addressing the stability of MB in differential experimental conditions. Another important aspect that needs more attention is the involvement of ultrasound and MB in triggering an immune response. Few studies have reported immune response activation by the administration of immune-stimulating substances assisted by ultrasound-mediated MB destruction [155]. The combination of ultrasound and MB causes differentiation and maturation of dendritic cells that leads to stimulation of T lymphocytes causing activation of antitumour immunity [155,156,157]. Although these findings are captivating they have only been validated in very few preclinical models. More studies focusing on the utilization of ligands that are more specific to individual immune cells should be conducted to explore the effectiveness of ultrasound-MB-based immunotherapies. Recently, nanobubbles have broadened their scope in theranostic applications [158]. As the name suggests nanobubbles, their smaller size allows them the transverse easily through the bilayer membrane. Due to their nano size, they remain buoyant and do not collapse easily maintaining their stability for a longer period. Whereas MB on the other hand tend to gradually reduce in size ultimately collapsing and dissolving their interior gases within the water surroundings. More studies should incorporate nanobubbles with ultrasound for systemic drug and gene delivery. Another crucial point that requires future exploration is the abscopal effect of the combinatory treatment (USMB and radiotherapy and or chemoradiotherapy). The abscopal effect generated by localized radiation or chemoradiotherapy to stimulate systemic antitumour effects has been documented widely however to our knowledge no studies have been carried out with combined USMB treatments. Investigating more about the abscopal effect following this combined treatment might help in treating metastatic tumours in a better way.

## Figures and Tables

**Figure 1 ijms-23-04393-f001:**
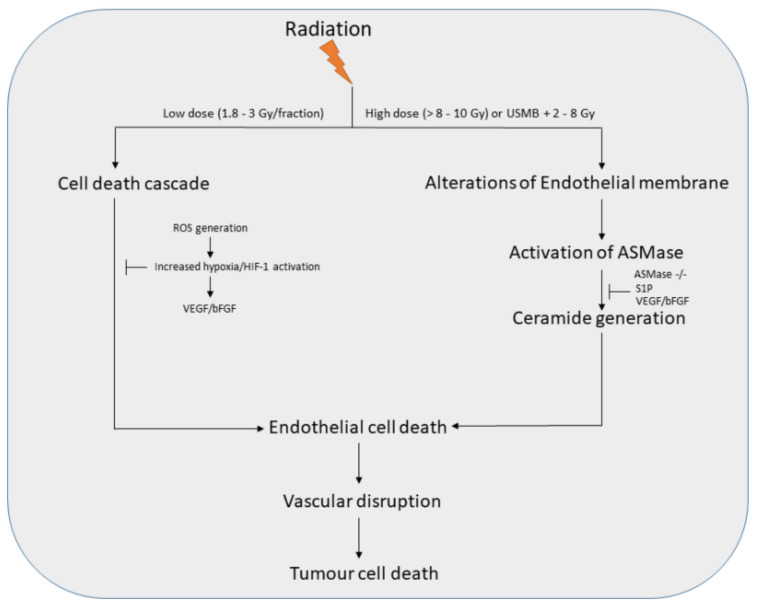
The low dose (1.8–3 Gy/fraction) induces the generation of ROS by hypoxia/reoxygenation that takes place following each exposure to radiation dose. This leads to the activation of HIF-1 which further generates several proangiogenic factors like VEGF or bFGF that debilitate radiation-induced endothelial cell apoptosis. However, inhibiting HIF-1 activity causes vascular radiosensitization resulting in tumour cell death. Exposure to a single high dose of radiation (>8–10 Gy) activates ASMase in endothelial cells to generate the secondary messenger ceramide. Ceramide accumulation in the plasma membrane facilitates apoptosis within endothelial cells, resulting in massive vascular disruption and subsequently leading to tumour cell death. USMB combined with low dose XRT elicits a similar effect as a single high dose. The addition of USMB to XRT is capable of activating ASMase/ceramide pathway resulting in endothelial cell apoptosis. This causes damage to tumour vasculature further leading to tumour cell death. Abbreviations: HIF-1 = hypoxia-inducible factor-1; bFGF = basic fibroblast growth factor; VEGF = vascular endothelial growth factor; USMB = ultrasound-stimulated microbubbles; XRT = radiotherapy.

**Figure 2 ijms-23-04393-f002:**
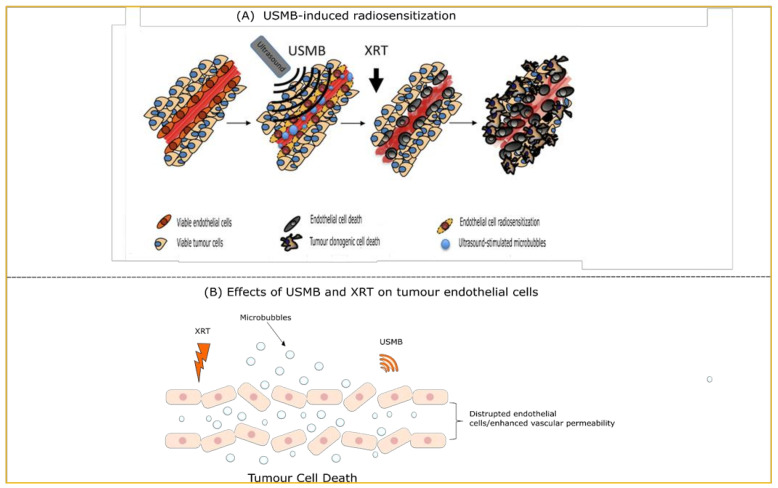
(**A**) Diagram illustrating the effect of USMB and XRT on tumour endothelial cells. Pretreatment of tumour cells with USMB radiosensitizes the cells. Exposing cells with XRT then cause endothelial cell damage followed by extensive tumour vasculature deterioration. This ultimately leads to tumour cell death. Administration of angiogenesis inhibitor can abrogate the entire process making tumour cells radioresistant. Adapted with permission from [73]. (**B**) USMB treatment results in endothelial cell membrane perturbation resulting in increased vascular permeability. The process of cavitation causes bubbles oscillation, contraction, and expansion finally causing it to collapse into tiny bubbles. This results in bioeffects of endothelial cells and nearby surrounding tissues. The treatment of USMB followed by XRT induces an enhanced tumour cell death/tumour response. Abbreviations: XRT = radiotherapy; USMB = ultrasound-stimulated microbubbles.

**Figure 3 ijms-23-04393-f003:**
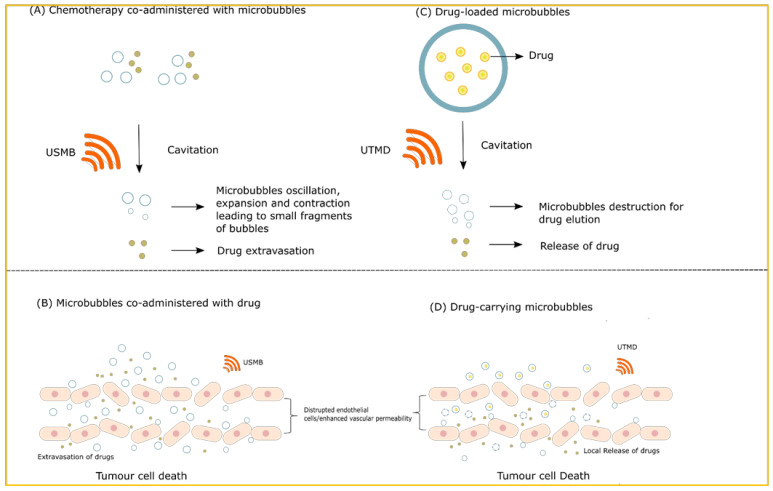
Illustration of drug delivery in conjugation with ultrasound and MB. (**A**) The drug can be co-injected with MB followed by ultrasound exposure to cause bubbles cavitation and its destruction leading to drug extravasation. (**B**) The image depicts the effect of co-administration of MB and drugs in ultrasound-guided drug delivery. USMB-induced sonoporation leads to increase vessel permeability allowing the uptake of drugs by the tumour cells causing them to destroy. (**C**) The drug can be delivered by loading them onto MB followed by an application of ultrasound acoustic pressure, the process known as ultrasound-targeted microbubble destruction (UTMD). The drug is eluted from MB causing its local release to the targeted site. (**D**) The image illustrates the effects of drug-loaded MB combined with ultrasound. UTMD enhances the efficacy of several drugs and their uptake by tumour cells through the effects of sonoporation. XRT = radiotherapy; USMB = ultrasound-stimulated microbubbles.

**Table 1 ijms-23-04393-t001:** USMB + XRT, chemotherapy-induced vascular effects observed in preclinical studies.

**(A) USMB + XRT**			
Treatment Type	Tumour Model	Tumour Vascular Effects	References
USMB + 2 Gy or 8 Gy	Mouse (human prostate cancer PC3)	Reduced blood flow, reduced vessel density, increased cell death, reduced cell proliferation	[21]
USMB + 2 Gy or 8 Gy	Mouse (human bladder cancer HT-1376)	Reduced blood flow, increased cell death, vascular normalization, increased fibrosis	[24]
USMB + 2 Gy, 4 Gy, or 8 Gy	Human umbilical vein endothelial cells (HUVEC), acute myeloid leukemia cells (AML), murine fibrosarcoma cells (KHT-C), prostate cancer cells (PC3), breast cancer cells (MDA-MB-231) and astrocytes cells	Increased nuclear fragmentation, reduced endothelial cell survival	[68]
USMB + 2 Gy or 8 Gy	Mouse (human breast cancer MDA-MB-231)	Reduced blood flow, reduced vessel density, increased cell death, inhibited tumour growth	[74]
USMB + 2 Gy or 8 Gy	Mouse (human prostate cancer PC3)	Reduced blood flow, reduced oxygen saturation, increased cell death, reduced vessel density	[75]
USMB + 2 Gy or 8 Gy	Mouse (human prostate cancer PC3)	Increased blood vessel leakage, reduced vessel density, increased hypoxia, increased cell death, reduced cell proliferation	[22]
USMB + 2 Gy or 8 Gy	Mouse (human prostate cancer PC3)	Increased cell disruption and cell death	[76]
USMB + 8 Gy	HUVEC cells	Increased cell death, reduced endothelial-cell tube formation	[72]
USMB + 2 Gy or 8 Gy	Mouse (human breast cancer MDA-MB-231)	Increased cell death, reduced vessel density, increased vascular leakage, inhibited tumour growth	[25]
USMB + 8 Gy	Mouse (human prostate cancer PC3)	Reduced blood flow, reduced oxygen saturation, increased cell death and fibrosis	[77]
USMB + 5 Gy	Rat (human hepatocellular carcinoma Hu7.5)	Reduced tumour vascularity, inhibited tumour growth	[78]
USMB + 2 Gy or 8 Gy	Human CNE-2 and HUVEC cells, Mouse (human CNE-2)	Reduced tumour cell viability, and formation of endothelial tubule, Reduced blood flow and CD34 expression, increased tumour cell death and increased ANG II and AT1R expression	[79]
USMB + 2 Gy or 8 Gy	Mouse (fibrosarcoma MCA/129)	Reduced blood flow and vessel density, increased cell death, inhibited tumour growth	[23]
USMB + 8 Gy	Mouse (human prostate cancer PC3)	Increased cell death, reduced vessel density	[80]
USMB + 8 Gy	Rabbit (human prostate cancer PC3)	Reduced blood flow, reduced oxygen saturation, increased cell death and fibrosis, reduced vessel density, inhibited tumour growth	[81]
USMB + 4 Gy	Human glioblastoma U87-MG cells, Mouse (human glioblastoma U87-MG)	Reduced CD34 expression, increased cell death, inhibited tumour growth	[82]
USMB + 2 Gy or 6 Gy	Human esophageal carcinoma cell lines (KYSE-510) and HUVEC cells	Reduced cell viability, reduced colony formation, increased cell death, inhibited angiogenesis, inhibited tumour growth, reduced cell proliferation	[83]
**(B) USMB/UTMD + Chemotherapy**			
**Treatment Type**	**Tumour Model**	**Tumour Vascular Effects**	**References**
USMB/UTMD + doxorubicin (DOX)	Rat (hepatocellular carcinoma 3924a)	Inhibited tumour growth	[84]
USMB + bevacizumab	Mouse (human 2LMP breast cancer)	Reduced tumour vascularity	[85]
USMB + docetaxel (DTX)	Mouse (human prostate cancer PC3)	Reduced tumour perfusion, increased cell death, inhibited tumour growth	[86]
USMB + DOX	Mouse (colorectal adenocarcinoma CT-26)	Disrupted tumour blood vessels, inhibited tumour growth	[87]
USMB + Metronomic cyclophosphamide (MCTX)	Mouse (human breast cancer MDA-MB-231)	Reduced tumour perfusion, increased cell death, inhibited tumour growth	[88]
USMB + paclitaxel (PTX)	Mouse (MIA PaCa- 2 luc)	Reduced tumour volume, sustained tumour vascularisation	[89]
USMB + DOX	Rat (9L gliosarcoma)	Increased K_trans_, vessel damage	[27]
USMB + DOX	Mouse (4T1 breast cancer)	Reduced tumour blood perfusion, increased levels of ROS, inhibited tumour growth, increased cell death	[90]
USMB + DOX	Rabbit (VX2 tumour)	Increased tumour perfusion, disrupted tumour microvessels, Inhibited tumour growth	[91]
USMB + DOX	Rabbit (VX2 tumour)	Increased vascular clearance of particles, reduced interstitial fluid pressure (IFP)	[92]
USMB + DOX	Mouse (neuroblastoma)	Increased tumour vascular permeability, reduced pericyte coverage, increased cell death	[93]
USMB + DOX	Mouse (human pancreatic carcinoma PANC-1)	Increased tumour blood perfusion	[94]

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
