# Peer review of "Application of Ultrasound Combined with Microbubbles for Cancer Therapy"

_ijms, 2022, doi:10.3390/ijms23084393_

Round 1

Reviewer 1 Report

The article "The Application of Ultrasound-Stimulated Microbubbles for Cancer Therapy" brings together the recent cancer therapy advancements by discussing the combined therapy of the Ultrasound-stimulated microbubbles (USMB). 

The manuscript has been well written, and the architecture of the paper made it easy to follow. Here are some comments: 

1- At first glance, there is too much text. Please remember that review papers are one of the educational sources, and it is always recommended to reduce the text and add more illustrations. You need to add a couple of more figures (up to 5 informative figures). The main purpose is to reduce the text without jeopardising the information provided. For instance, in line 195, "Many studies have explored the enhancing effects..." you may compare the details of the studies such as samples, gender, type of cancer, duration, etc. The same idea for line 251, "When combining two different treatment...", which is a critical section in the paper.  

2- Every molecular scientist wants to see in vitro studies; therefore, please attempt to illustrate those data by paying attention to the type of sample, cell line, animals, method, etc. 

3- Please also do the same for in vivo preclinical and clinical studies. These are for different interest groups. 

A suggestion for future papers: It's really tempting to see whether is any meta-analysis in this area (combine therapy using USMB). You may want to be the first group to work on that. So exciting to see what statistics is trying to tell us about the USMB combined therapy.

PS: The order of figures is weird; I assume something has happened while dragging them to the text. Please correct that. 

Stay safe in Toronto. 

Author Response

We thank the editors and reviewers for their comments and constructive criticism. We have incorporated responses to their critique on all points throughout the manuscript. We believe that this has helped strengthen our review article and has improved the overall science presented in the review manuscript.

Reviewer 1

Comments and Suggestions for Authors

The article "The Application of Ultrasound-Stimulated Microbubbles for Cancer Therapy" brings together the recent cancer therapy advancements by discussing the combined therapy of the Ultrasound-stimulated microbubbles (USMB).

The manuscript has been well written, and the architecture of the paper made it easy to follow. Here are some comments:

Reviewer Comment

1- At first glance, there is too much text. Please remember that review papers are one of the educational sources, and it is always recommended to reduce the text and add more illustrations. You need to add a couple of more figures (up to 5 informative figures). The main purpose is to reduce the text without jeopardising the information provided. For instance, in line 195, "Many studies have explored the enhancing effects..." you may compare the details of the studies such as samples, gender, type of cancer, duration, etc. The same idea for line 251, "When combining two different treatment...", which is a critical section in the paper.

Author reply: We thank the reviewer for such a positive output for our review article. We would like to point out that in this review article we tried compiling mostly data that was related to USMB-induced tumour vascular effects. However, we have now added additional studies discussing USMB induced radiosensitization. Regarding the figures, since our lab has several original figures regarding USMB+XRT however for USMB+chemotherapy we need permission for it. We have now added tables as suggested by the reviewers.

Reviewer Comment

2- Every molecular scientist wants to see in vitro studies; therefore, please attempt to illustrate those data by paying attention to the type of sample, cell line, animals, method, etc.

Author reply: We have now added tables as suggested by the reviewers. We have also added additional studies in the section of USMB+radiotherapy.

Reviewer Comment

3- Please also do the same for in vivo preclinical and clinical studies. These are for different interest groups.

Author reply: We have now added tables as suggested by the reviewers. As per our knowledge, there aren’t any clinical studies so far that reported tumour vascular effects.

Reviewer Comment

A suggestion for future papers: It's really tempting to see whether is any meta-analysis in this area (combine therapy using USMB). You may want to be the first group to work on that. So exciting to see what statistics is trying to tell us about the USMB combined therapy.

PS: The order of figures is weird; I assume something has happened while dragging them to the text. Please correct that.

Author reply: To our knowledge, there aren't any meta-analysis studies so far dealing with USMB and cancer therapies.

The order of the figure has been changed during the submission process. We have now arranged every figure in an orderly manner.

Stay safe in Toronto.

Thank you.

Reviewer 2 Report

Ijms-1639687 review

This paper presents a review of the interest of Ultrasound-stimulated microbubbles in cancer treatment. It summarizes current knowledge of mechanism of action and various applications of the technique. It then lists studies addressing the potential interest of the association of USMB with either radiotherapy or chemotherapy. Next, it concentrates on the use of USMB to deliver drugs to cancer sites or to disrupt the blood-brain barrier. There is also a small part concerning studies of USMB in spheroid cultures. Finally, it recapitulates the first clinical trials using this technology.

The review is interesting but needs improvement.

There is a lot of self-citation, especially in the introduction and part 2.

Another general point: It needs to be specified when results are in model organisms: as in line 62, 81, 195, 330, and line 516 to 517.

Concerning the part “2.USMB and radiotherapy”:

It needs to be improved to be more readable. It is a list of studies and the interest of each one compared to the other is not pointed at. It would need to be more synthetic or at least give an idea what each study brings to the model or hypothesis and end up to a generalization of the message.

Despite the fact that “many studies have explored the enhancing effects of USMB on a variety of cancer types” (line 195) the part lists only the studies conducted by the laboratory writing the review, this is odd. Has nobody else worked on the subject? This should be rebalanced.

Part 5:

It needs to be re-written. It starts with explaining how “3D cell culture systems developed using patient-derived organoids or patient-derived xenograft” (line 526) could provide a more realistic model. But then, it presents studies done on spheroids formed from cell lines. There is a confusion on models which are very different. The interest of spheroids compared to 2D culture exists, but is should be clarified, and is different from the interest of organoid (derived from patient biopsies). It would be interesting to discuss also the fact that USMB action is supposed to be mediated by the vascular system that is absent from both models (which is a drawback compared to animal studies). All that should be clarified and better discussed.

Finally, references are needed in:

Line 40,

line 65

line 81

paragraph 576 to 591

Sentences 516 to 518

Minor points:

Figures are not is the right order.

Figure 3 legend is missing (C and D not described).

If there is any indication of how USMB induces radiosensibilization, it would be of interest to mention it.

Author Response

We thank the editors and reviewers for their comments and constructive criticism. We have incorporated responses to their critique on all points throughout the manuscript. We believe that this has helped strengthen our review article and has improved the overall science presented in the review manuscript.

Reviewer 2

Comments and Suggestions for Authors

Ijms-1639687 review

This paper presents a review of the interest of Ultrasound-stimulated microbubbles in cancer treatment. It summarizes current knowledge of mechanism of action and various applications of the technique. It then lists studies addressing the potential interest of the association of USMB with either radiotherapy or chemotherapy. Next, it concentrates on the use of USMB to deliver drugs to cancer sites or to disrupt the blood-brain barrier. There is also a small part concerning studies of USMB in spheroid cultures. Finally, it recapitulates the first clinical trials using this technology.

The review is interesting but needs improvement.

Reviewer Comment

There is a lot of self-citation, especially in the introduction and part 2.

Author reply: We agree with the reviewer that the paper contains self-citation. We wanted to focus the review mainly on discussing the mechanism behind USMB and XRT, chemotherapy-related tumour vasculature damage. This topic has mainly been explored by our group. However, we have now included more studies discussing the combined effect of USMB and radiotherapy.

There are several effects observed following USMB+XRT, chemotherapy apart of vascular effects however the review will be too long if we even intend to attempt it.

Reviewer Comment

Another general point: It needs to be specified when results are in model organisms: as in line 62, 81, 195, 330, and line 516 to 517.

Author reply: We have now added mentioned that the studies were performed in an in vitro and in vivo or preclinical models. We have explained each and every study by specifying the cells or xenograft.

Reviewer Comment

Concerning the part “2.USMB and radiotherapy”:

It needs to be improved to be more readable. It is a list of studies and the interest of each one compared to the other is not pointed at. It would need to be more synthetic or at least give an idea what each study brings to the model or hypothesis and end up to a generalization of the message.

Despite the fact that “many studies have explored the enhancing effects of USMB on a variety of cancer types” (line 195) the part lists only the studies conducted by the laboratory writing the review, this is odd. Has nobody else worked on the subject? This should be rebalanced.

Author reply: We thank the reviewer for raising this important concern. However, in our present review, we only wanted to discuss the effect of USMB and XRT within tumor vasculature. However, we have now added more studies discussing USMB induced radiosensitization effect in the USMB+radiotherapy section.

Reviewer Comment

Part 5:

It needs to be re-written. It starts with explaining how “3D cell culture systems developed using patient-derived organoids or patient-derived xenograft” (line 526) could provide a more realistic model. But then, it presents studies done on spheroids formed from cell lines. There is a confusion on models which are very different. The interest of spheroids compared to 2D culture exists, but is should be clarified, and is different from the interest of organoid (derived from patient biopsies). It would be interesting to discuss also the fact that USMB action is supposed to be mediated by the vascular system that is absent from both models (which is a drawback compared to animal studies). All that should be clarified and better discussed.

Finally, references are needed in:

Line 40,

line 65

line 81

paragraph 576 to 591

Sentences 516 to 518

Author reply: We agree with the reviewer regarding the confusion between spheroids and organoids or patient-derived xenografts. In this section, we aim to explain the importance of spheroids for studying the efficacy of USMB and chemotherapy. We wanted to explain a few studies that succeeded to demonstrate the combinatorial effect of this combined therapy.

We have now deleted the line that mentioned organoids or patient-derived xenograft which was not relevant to this part.

We apologize for overlooking the reference part. We have now added references to the manuscript where required.

Reviewer Comment

Minor points:

Figures are not is the right order.

Author reply: The order of the figure was changed during the submission process by mistake however every figure is now in order.

Reviewer Comment

Figure 3 legend is missing (C and D not described).

Author reply: We thank the reviewer for pointing this out. We have now added descriptions for Figure 3 C and D legend.

Reviewer Comment

If there is any indication of how USMB induces radiosensibilization, it would be of interest to mention it.

Author reply: We have now mentioned some studies that discussed USMB-induced radiosensitization in our revised manuscript (section: USMB and radiotherapy).

Reviewer 3 Report

Sharma et al. made a comprehensive review about ultrasound-stimulated microbubbles (USMB) for cancer therapy.  The review provides a strong summary of the field.  The authors discussed successful data well but did not for limitations.  For example, 1) how would the immune systems play a role in USMB + radiotherapy/chemotherapy, particularly in in vivo studies?; 2) How would the abscopal effect contribute to USMB and radiotherapy?; 3) In section 4.2, what would be the advantage and disadvantage of drug-loaded microbubbles compared with nanoparticle-mediated drug delivery? Those issues can be further addressed.   

Author Response

We thank the editors and reviewers for their comments and constructive criticism. We have incorporated responses to their critique on all points throughout the manuscript. We believe that this has helped strengthen our review article and has improved the overall science presented in the review manuscript.

Reviewer 3

Comments and Suggestions for Authors

Reviewer Comment

Sharma et al. made a comprehensive review about ultrasound-stimulated microbubbles (USMB) for cancer therapy.  The review provides a strong summary of the field.  The authors discussed successful data well but did not for limitations.  For example, 1) how would the immune systems play a role in USMB + radiotherapy/chemotherapy, particularly in in vivo studies?; 2) How would the abscopal effect contribute to USMB and radiotherapy?; 3) In section 4.2, what would be the advantage and disadvantage of drug-loaded microbubbles compared with nanoparticle-mediated drug delivery? Those issues can be further addressed.

Author reply: We thank the reviewer for pointing out the limitation part. We have now included a limitation part where we have discussed all the mentioned suggestions by the reviewer.

Round 2

Reviewer 2 Report

Thank you to the authors who have considered in the revised version of the manuscript most important points I raised. In my opinion the combination of all modifications has greatly improved the quality of the paper.

I have just three minor points to raise:

Line 241: a "ref" has not been replaced by the corresponding number.

Line 296: the reference for the work of He and al is missing.

Line 864: One cannot speak of abscopal effect for chemotherapeutic drugs

Author Response

We thank the editors and reviewers for their comments and constructive criticism. We have incorporated responses to their critique on all points throughout the manuscript. We believe that this has helped strengthen our review article and has improved the overall science presented in the review manuscript.

Reviewer 2

Reviewer Comment

Line 241: a "ref" has not been replaced by the corresponding number.

Author reply: The word ref has now been replaced by the actual references.

Line 296: the reference for the work of He and al is missing.

Author reply: The work of He et al has now been referenced and cited in the manuscript.

Line 864: One cannot speak of abscopal effect for chemotherapeutic drugs

Author reply: The word chemotherapeutic drugs have now been replaced by chemoradiotherapy.
